# Feasibility study of a novel technique for treating refractory ascites using a peritoneobladder shunt in a swine model

**Byeong Hwa Bak**[1☯], **Jung Hyuk Ko**[2☯], **Myoung Joo Kang**[3‡], **Yong June Lee**[3‡], **Ki Won Kim**[4‡], **Joo Yeon Kim**[5‡], **Jung Hoon Kim**[6‡], **Jae Joon Kim**[7‡], **Seok Jae Huh**[8‡], **Cheol Kyu Oh**[9‡], **Il Hwan Kim**[10]*

1 Devision of Nephrology, Hanbithospital, Busan, Republic of Korea, 2 Department of Mechanical Engineering, Korea Maritime & Ocean University, Busan, Republic of Korea, 3 Division of Oncology, Department of Internal Medicine, Haeundae Paik Hospital, Inje University College of Medicine, Busan, Republic of Korea, 4 Research and Development Department, Wabetree. Co. Ltd., Gimhae-si, Republic of Korea, 5 Department of Pathology, Inje University, College of Medicine, Haeundae Paik Hospital, Busan, Republic of Korea, 6 Division of Hematology/Oncology, Department of Internal Medicine, Pusan National University Yangsan Hospital, Busan, Republic of Korea, 7 Division of Hematology & Oncology, Department of Internal Medicine, Pusan National University Yangsan Hospital, Busan, Republic of Korea, 8 Department of Internal Medicine, Dong-A University College of Medicine, Busan, Republic of Korea, 9 Department of Urology, Haeundae Paik Hospital, Busan, Republic of Korea, 10 Department of Internal Medicine, Division of Oncology, Inje University College of Medicine, Haeundae Paik Hospital, Cancer Center, Busan, Republic of Korea

☯ These first authors contributed equally to this work.
‡ These coauthors contributed equally to this work.
* onelement@naver.com

## Abstract

Ascites is often treated through paracentesis, which requires repeated application and poses risks. In this study, we developed a peritoneobladder shunt that allows natural ascites drainage from the peritoneal cavity to the bladder. We conducted an experiment to determine the functional effect of the peritoneobladder shunt in a swine model. Peritoneobladder shunts were developed and placed in 4 swine models to test their effectiveness in draining ascitic fluid from the peritoneal cavity to the bladder. The peritoneobladder shunts were inserted laparoscopically; some models received the shunts with one-way check valves to prevent fluid reflux, while one received peritoneobladder shunts without the valves. After a short (7 days) survival period, experiments were conducted to verify that the peritoneobladder shunts were properly fixed in the bladder. During the survival period, two peritoneobladder shunts could be placed in the bladder wall without rupture or tearing. When the peritoneal cavity was filled with an ascitic fluid substitute, the fluid naturally drained into the bladder; when the peritoneobladder shunts with one-way check valves were used, no reflux occurred from the bladder into the peritoneal cavity. In experiments using swine models, the peritoneobladder shunts effectively drained fluid, and one-way check valves successfully prevented reflux. These findings suggest that the peritoneobladder shunt

**Data availability statement:** All relevant data is included in the paper and attached figures.

**Funding:** This research was supported by a grant of the Korea Health Technology R&D Project through the Korea Health Industry Development Institute (KHIDI) funded by the Ministry of Health & Welfare, Republic of Korea (grant number: RS-2024-00407339).

**Competing interests:** The author declared that there are no competing interests.

could be an alternative option to alleviate the burden on patients who require paracentesis, allowing home-based treatment. Further studies are needed to assess the long-term stability and safety of this procedure in humans.

## Introduction

### Ascites is caused by various diseases and can significantly reduce quality of life, making repeated treatment and management challenging

Ascites, the accumulation of fluid in the peritoneal cavity can result from various conditions, such as cirrhosis, malignancies, tuberculous peritonitis, and kidney and heart disease. [1,2] Depending on the underlying cause, it can be classified as exudative or transudative. [3] In patients with compensated cirrhosis, ascites occurs at an annual rate of 5–10% and in approximately 50% of patients with primary tumors, indicating a very high prevalence. [1,4,5]

Fluid accumulation in the peritoneal cavity can lead to abdominal distention, which may restrict dietary intake and result in malnutrition. [6,7] It can also cause complications such as dyspnea, impaired mobility, pleural effusion, hernia, and systemic edema. [8,9] Proper management is crucial for preventing or addressing these issues; available options include medication, paracentesis, catheter insertion, and surgery. [2,10] Among these methods, paracentesis is frequently used because of its simplicity and quick execution. [2,11] However, because ascites often recurs, the procedure must be repeated, increasing the risk of secondary problems such as bleeding, perforation, and infection. [12,13] Each session is also associated with pain from needle insertion. Additionally, a caregiver must accompany the patient and wait for the procedure to finish, which can interfere with their daily and professional responsibilities. [14,15]

### A peritoneobladder shunt was designed for home-based ascites management by allowing natural fluid drainage into the bladder

To address these issues, we developed a peritoneobladder shunt, a device that enables home-based ascites management by allowing fluid to drain naturally from the peritoneal cavity into the bladder. [16] Under normal physiological conditions, the intraabdominal pressure and bladder pressure are equal; thus, no fluid movement occurs through the periotneobladder shunt. [16] However, when the ascites fluid accumulates, the intraabdominal pressure exceeds the bladder pressure, allowing drainage through the peritoneobladder shunt. We hypothesized that in a swine model with artificially induced ascitic fluid, connecting the peritoneal cavity to the bladder via a peritoneobladder shunt would result in spontaneous fluid transfer.

The peritoneobladder shunt is the device we developed for this experiment. Diagrams of the peritoneobladder shunt, its position in the body, and the resulting drainage are shown in Fig 1. We also investigated the performance of one-way check valves by implanting peritoneobladder shunts with and without these valves.

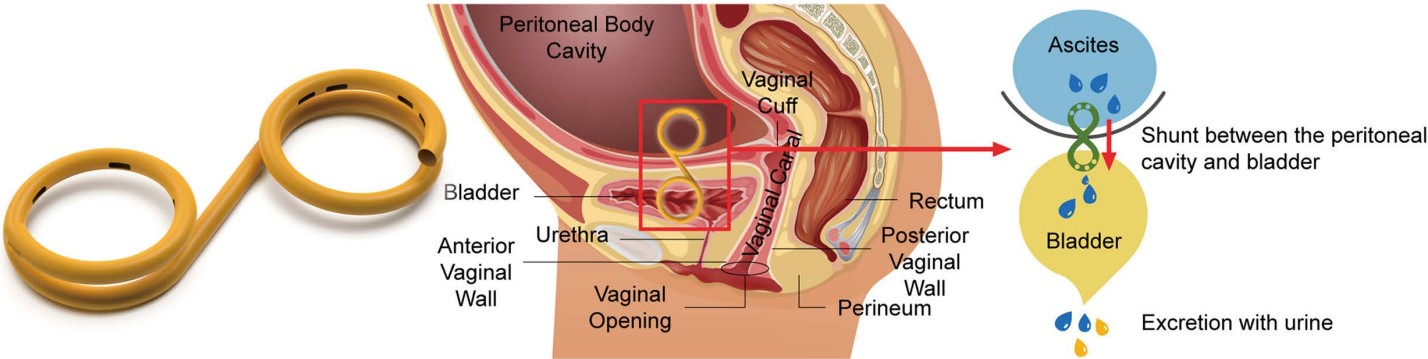

**Fig 1. Diagrams of the peritoneobladder shunt, its position in the body, and drainage.**

We aimed to test the following hypotheses to determine the clinical suitability of the periotneobladder shunt:

1. Elevated peritoneal pressure allows ascitic fluid to drain naturally into the bladder through peritoneobladder shunts inserted in the bladder wall.

2. Oe-way check valves in the peritoneobladder shunt prevent reflux of urine or ascitic fluid from the bladder back into the peritoneal cavity.

3. The peritoneobladder shunt remains securely fixed in the bladder wall without migrating into the peritoneal cavity or bladder because of bladder movement or patient activity. Displacement into the peritoneal cavity would irritate the peritoneal membrane, potentially causing bowel paralysis, enteritis, perforation, or other complications, and may require surgical removal.

To test these three hypotheses, we implanted peritoneobladder shunts in a swine model and evaluated short-term survival over 7 days. A subsequent dismantling experiment confirmed whether the peritoneobladder shunts remained properly fixed in the bladder wall.

We aimed to demonstrate the feasibility of peritoneobladder shunt interval drainage for ascites. To our knowledge, this is the first study exploring methods to enable internal drainage within the body. This technique holds significant potential to improve the daily lives of both patients and caregivers.

## Materials and methods

All procedures were approved by the Animal Experimentation Ethics Committee of Pusan National University Hospital (approval number: 2021–046-A1C0(0)). The institutional and/or licensing committees approved the experiments, including any relevant details, confirming that all the experiments were performed in accordance with the relevant guidelines and regulations.

This experiment was conducted with reference to the ARRIVE guidelines 2.0. [17].

### Materials

**Experimental device.** The name, manufacturer, and product code of the devices used in this animal experiment are summarized in Table 1. The peritoneobladder shunts were specifically developed for this study and featured a double-ring structure: one ring in the peritoneal cavity and the other ring in the bladder; this structure prevents reflux of ascitic fluid from the bladder back into the peritoneal cavity. The ring located in the peritoneal cavity contains multiple one-way check valves. These valves have a thin membrane structure that, under external pressure, prevents reflux of fluid already

**Table 1. Names, Manufacturers, and Product codes of the experimental devices.**

| Device name | Manufacturer | Product code |
| --- | --- | --- |
| Percutanous drainage 14 French | Sungwon Medical Co., Ltd. | PI1420-RLC (30) |
| Dilator | Wabetree. Co. Ltd | No code (self-developed) |
| Dilator | Bard | 231135 |
| Guide wire | Bard | 145FS38 |
| Guide wire | Bard | 145FS35 |
| Guide wire | Wabetree. Co. Ltd | No code (self-developed) |
| Foley Catheter 8 French | Sewoon Medical Co., Ltd. | 2101−008 |
| Trocar diameter 12 mm | Ethicon | B12LT |
| Laparoscope | Olympus | Hopkins ii–26003AA |
| Pusher | Wabetree. Co. Ltd | No code (self-developed) |
| Peritoneobladder shunt (without one-way check-valves) | Wabetree. Co. Ltd | No code (self-developed) |
| Peritoneobladder shunt (with one-way check-valves) | Wabetree. Co. Ltd | No code (self-developed) |

drained into the bladder. To evaluate their performance, experiments were conducted on peritoneobladder shunts both with and without the check valves. Details of the peritoneobladder shunt are shown in Figs 2 and 3.

**Artificial ascitic fluid substitute.** The ascitic fluid, inserted via percutaneous drainage (PCD), was a mixture of normal saline with methylene blue at a 100:1 ratio (10 cc of methylene blue in 1 L of normal saline).

**Sedatives and anesthetics.** Thirty minutes before the start of the procedure, we administered two doses of alfaxalone (20 mg) and two doses of xylazine (10 mg) to the swine model intramuscularly as analgesia. Moreover, one dose of tramadol (50 mg) was administered intravenously to relieve pain.

After the procedure was complete and before transfer to the animal facility, one dose of tramadol (50 mg) was also administered intravenously to relieve pain. Following transfer to the animal facility, 50 mg tramadol was administered twice daily intramuscularly.

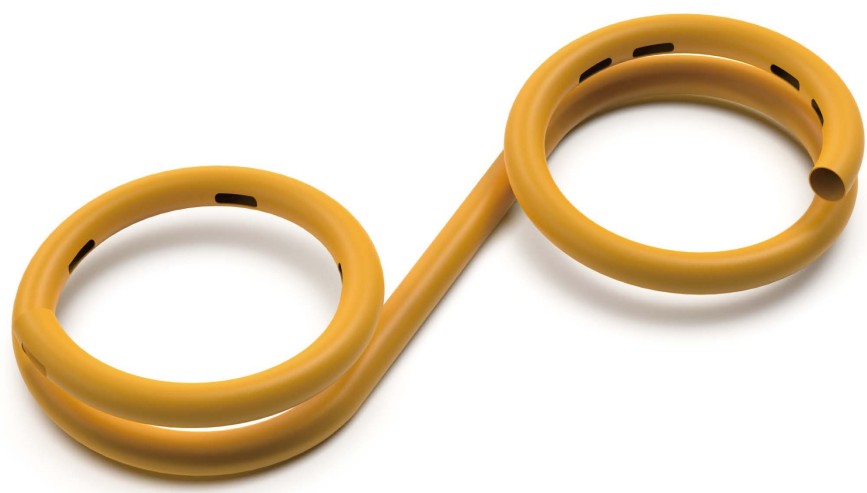

**Fig 2. Peritoneobladder shunt without one-way checkvalves.**

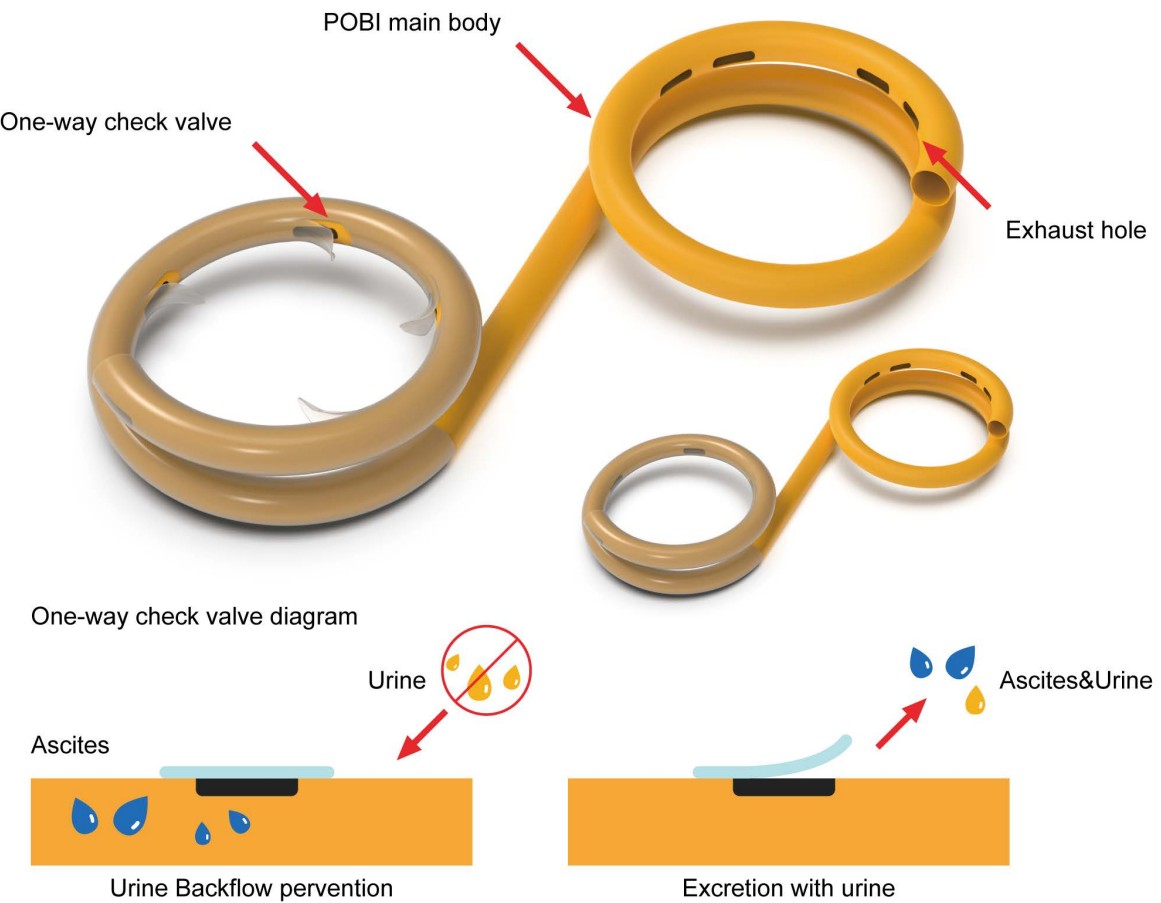

**Fig 3. Peritoneobladder shunt with one-way check valves and Principle of a one-way check valve in a peritoneobladder shunt].**

In the dismantling study, 30 minutes before the start of the procedure, we administered two doses of 20 mg of alfax-alone and 10 mg of xylazine to the swine model intramuscularly as analgesia. Moreover, one dose of tramadol (50 mg) was administered intravenously to relieve pain.

**Disinfection.** The peritoneobladder shunt was disinfected using ethylene oxide gas, the methylene blue and methylene red stains were disinfected using gamma radiation, and the guidewires were disinfected using povidone.

**Experimental animals.** Four healthy pigs with no underlying conditions were used in this experiment. The pigs used in the experiment were initially raised at the Minjeong Livestock animal facility and then underwent a two-week quarantine and acclimatization period at the Preclinical Trial Center of Pusan National University Yangsan Hospital. Table 2 provides detailed information on their age, sex, weight, origin, diet and feeding schedule, housing conditions, and any preexisting health conditions.

## Methods

### Experimental preparation.

(1) Thirty minutes before the start of the procedure, each pig was anesthetized with two doses of alfaxalone and xylazine intramuscularly, along with a single dose of tramadol intravenously to relieve pain.

**Table 2. Age, sex, weight, feed and feeding schedule, rearing environment, and underlying disease in the swine model.**

| | Swine model 1 | Swine model 2 | Swine model 3 | Swine model 4 |
|---|---|---|---|---|
| **Age/Sex** | 12 weeks/F | 12 weeks/F | 12 weeks/F | 12 weeks/F |
| **Weight** | 42 kg | 45 kg | 42 kg | 40 kg |
| **Rearing location** | Minjeong Livestock Farm (173, Myeongdong-ro, Hallim-myeon, Kimhae-si, Gyeongsangnam-do). | | | |
| **Feed and feeding schedule** | Swine feed (Purina Lintek Plus 90) | | | |
| | Feeding twice per day, 600 g each time | | | |
| **Rearing environment** | Temperature 21–26 °C | | | |
| | Humidity 30–70% | | | |
| | Light-dark cycle 12 hrs | | | |
| | Automated water supply | | | |
| | Air ventilation rate 10–15 times/hr | | | |
| **Underlying disease** | None | | | |

(2) After confirming that the pig was anesthetized, it was secured to the operating table in the Trendelenburg position.

(3) The surgical site and surrounding area were covered with 7.5% povidone sterile drapes.

(4) The rest of the pig was covered with sterilized drapes to avoid contamination.

**PCD of the peritoneal cavity.**

(1) After an incision was made in the superior left quadrant of the abdomen, a trocar was inserted, and $CO_2$ was injected.

(2) A laparoscope was inserted via the trocar to verify the location of the bladder, and a 14 French PCD was placed in the right abdomen.

**Peritoneobladder shunt placement in the bladder of the swine model.**

(1) After an 8 French Foley catheter was inserted, the bladder was emptied, and a urine bag was connected.

(2) Then, 300 mL of normal saline was injected via the Foley catheter to stretch the bladder to make the shunt placement procedure easier to perform.

(3) A laparoscope was inserted via the trocar in the superior left abdomen.

(4) Using a laparoscopic needle, the bladder was punctured from the peritoneal cavity, and a guidewire was inserted.

(5) A dilator was inserted via the guidewire to dilate the puncture site.

(6) After the dilator was removed, the peritoneobladder shunt was advanced along the guidewire from the peritoneal cavity to the bladder.

(7) After the peritoneobladder shunt was placed at the bladder puncture site, it was pushed using a pusher. Swine model 1 received a peritoneobladder shunt without one-way check valves, whereas swine models 2–4 received peritoneobladder shunts with one-way check valves. Steps (3) to (7) were repeated for the second peritoneobladder shunt.

(8) The pusher and guidewire were removed.

(9) The PCD catheter placed in the peritoneal cavity was used to inject 3 L of the normal saline/methylene blue mixture into the peritoneal cavity.

(10) The movement of the diluted methylene blue solution from the peritoneal cavity to the bladder through the peritoneobladder shunt was confirmed immediately by the excretion of the saline/methylene blue mixture through the Foley catheter.

(11) To check whether there was reflux of bodily fluids (urine + ascitic fluid) from the bladder to the peritoneal cavity via the peritoneobladder shunt, 100 mL of methylene red was mixed with 1 L of normal saline, and 300 mL of this mixture was injected into the Foley catheter.

(12) The laparoscope was used to visually check for any reflux of methylene red from the bladder to the peritoneal cavity.

(13) The laparoscope was removed, and the site of insertion was sutured.

(14) To prevent potential bladder injury caused by accidental tearing if the animal awoke with the Foley catheter still in place, it was removed prior to recovery from anesthesia.

(15) After the experiment was completed and once the pig recovered from anesthesia, 50 mg of tramadol was administered intravenously 30 minutes before it was sent to the animal facility. Following transfer to the animal facility, 50 mg tramadol was administered twice daily intramuscularly. The condition of swine models 1~4 was monitored, including observations of any signs of inflammation, such as fever or chills, by measuring core temperature (ex. rectal thermometer). In addition, to confirm whether the ascitic fluid was discharged into the bladder, the presence of methylene blue solution in the pens was visually inspected twice daily.

**Verification of performance and specimen analysis after the procedure.** After evaluating the short-term (7 days) survival of the models, we performed a dismantling experiment to verify that the peritoneobladder shunt was properly fixed in the bladder.

**Experimental preparation:**

① Thirty minutes before the start of the procedure, we administered analgesia; the swine model was anesthetized using two doses of 20 mg of alfaxalone and two doses of xylazine (10 mg per animal) intramuscularly. Moreover, one dose of tramadol (50 mg) was administered intravenously to relieve pain.

② After confirming that the pig was anesthetized, it was secured to the operating table in the Trendelenburg position.

③ The surgical site and surrounding area were covered with 7.5% povidone sterile drapes.

④ The rest of the pig was covered with sterilized drapes to avoid contamination.

**Dismantling experiment:**

① After confirming that the pig was anesthetized, 40 mg of KCl was administered intravenously, and cardiac arrest was verified.

② The bladder was isolated, and the peritoneobladder shunt was check to confirm whether it was still well placed within the bladder.

## Results

This experiment was conducted over a short-term survival period of 7 days. During this time, we confirmed that the peritoneobladder shunt was properly inserted in swine models 1–4. Ascitic fluid was effectively drained from the peritoneal cavity to the bladder through the peritoneobladder shunt. The one-way check valves were fixed in the peritoneobladder shunt, effectively preventing reflux from the bladder back into the peritoneal cavity. In the dismantling experiments performed after 7 days, we verified whether the peritoneobladder shunts remained fixed in the bladder wall.

## Peritoneobladder shunt in swine model 1

In swine model 1, the peritoneobladder shunt was easily inserted into the bladder, and we confirmed that the two peritoneobladder shunts could be placed simultaneously (Fig 4A). Laparoscopic observation confirmed that the mixture of normal saline and methylene blue was rapidly excreted through the Foley catheter. Additionally, laparoscopic visualization confirmed reflux of the mixture of normal saline and methylene red from the bladder back into the peritoneal cavity.

After the swine model was returned to the animal facility, during the 7-day observation period, we visually confirmed that methylene blue dye was excreted in the urine and stained the animal's skin, such as on the legs and hips. Throughout this period, the core temperature was measured three times a day, which revealed no signs of fever or infection, and the food intake remained consistent with the previous levels, with feeding twice per day (600 g each). After 7 days, the dismantling experiment revealed that one peritoneobladder shunt remained properly fixed in the bladder wall, whereas the other had become detached and was found inside the bladder (Fig 4B).

## Peritoneobladder shunt in swine models 2–4

In swine models 2–4, we used peritoneobladder shunts with one-way check valves. We confirmed that two peritoneobladder shunts could be placed simultaneously. Laparoscopic observation confirmed that the mixture of normal saline and methylene blue was rapidly excreted through the Foley catheter. Additionally, laparoscopic visualization confirmed the absence of reflux of the mixture of normal saline and methylene red from the bladder back into the peritoneal cavity (Fig 5A).

After the pigs were returned to the animal facility, during the 7-day observation period, we visually confirmed that methylene blue dye was excreted in the urine and stained the animal's skin, such as on the legs and hips. Throughout this period, the core temperature was measured three times a day, which revealed no signs of fever or infection, and the food intake remained consistent with the previous levels, with feeding twice per day (600 g each time). After 7 days, the dismantling experiments showed that the two peritoneobladder shunts remained properly fixed to the bladder wall (Fig 5B–5D).

The experimental results for swine models 1–4 are summarized in Table 3.

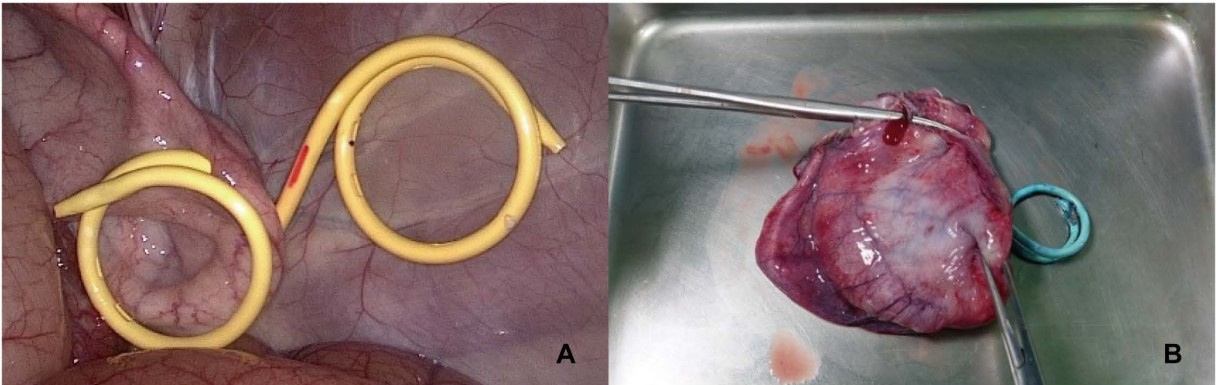

**Fig 4. A** Two peritoneobladder shunts placed in the bladder of swine model 1, confirmed laparoscopically. **B** One peritoneobladder shunt was confirmed to be situated in the bladder wall of swine model 1 in the dismantling experiment.

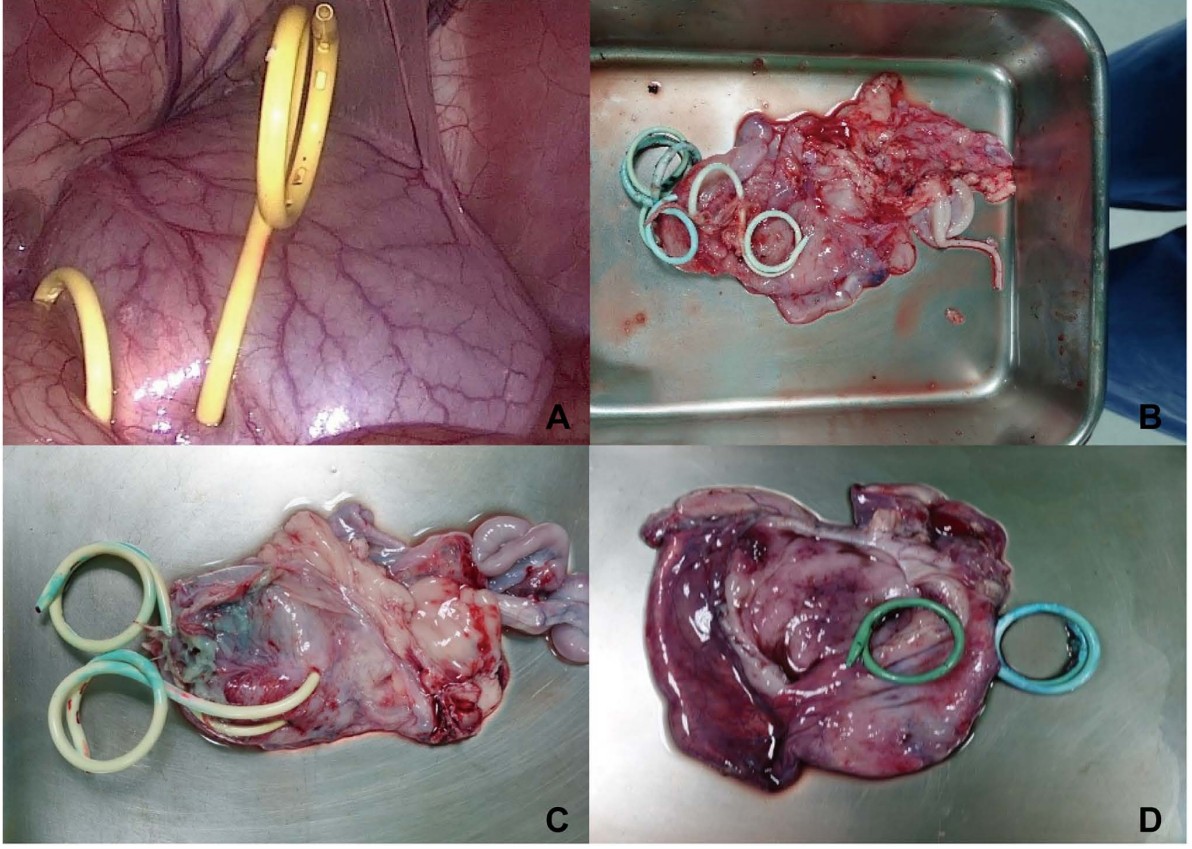

**Fig 5. A** Two peritoneobladder shunts placed in the bladder of swine model 2 without reflux from the bladder back into the peritoneal cavity, confirmed laparoscopically **B** Confirmation of two peritoneobladder shunts placed in the bladder wall of swine model 2 in the dismantling experiment **C** Confirmation of two peritoneobladder shunts placed in the bladder wall of swine model 3 in the dismantling experiment **D** Confirmation of two peritoneobladder shunts placed in the bladder wall of swine model 4 in the dismantling experiment.

## Discussion

### Peritoneobladder shunts can be safely placed in the bladder wall to enable natural, one-way drainage of ascitic fluid, suggesting a potential alternative to paracentesis for patients

In this study, we confirmed that during a 7-day short-term survival period, peritoneobladder shunts could be placed in the bladder wall. Even two peritoneobladder shunts could be placed in the same swine model without rupture or tearing of the bladder wall. When the peritoneal cavity was filled with an ascitic fluid substitute, natural drainage occurred into the bladder, and when a peritoneobladder shunt with one-way check valves was used, there was no reflux from the bladder into the peritoneal cavity. These findings suggest that the peritoneobladder shunt could be an alternative option to alleviate the burden on patients who require paracentesis.

We used a swine model as the experimental animal because the bladder conditions, such as capacity and cross-sectional area, are most similar to those of humans. [18] However, the anatomical placement of the urethra differs between the swine model and humans, which can make finding the urethral orifice difficult. To account for this, rather than using younger or smaller swine models, we used those aged at least 12 weeks and weighing at least 40 kg, making it easier to find the urethral orifice.

Table 3. **Experimental results for swine models 1–4: peritoneobladder shunt type, verification of peritoneobladder shunt placement, number of peritoneobladder shunts placed, verification of methylene blue excretion via the Foley catheter, verification of methylene red reflux from the bladder to the peritoneal cavity, and survival period.**

| Swine model | Peritoneobladder shunt type | Peritoneobladder shunt placement/count | Verification of methylene blue excretion via the Foley catheter | Verification of methylene red reflux from the bladder to the peritoneal cavity | Survival period |
|---|---|---|---|---|---|
| 1 | Check-valves absent | Placed/2 shunts | Excreted | Reflux observed | 7 days |
| 2 | Check-valves present | Placed/2 shunts | Excreted | No reflux observed | 7 days |
| 3 | Check-valves present | Placed/2 shunts | Excreted | No reflux observed | 7 days |
| 4 | Check-valves present | Placed/2 shunts | Excreted | No reflux observed | 7 days |

Owing to the anatomical structure of the swine model, male pigs could not be used in this experiment, and only female pigs were included. Since there are no anatomical differences in the bladder itself between sexes, the procedure outcomes are not expected to differ.

## Studies conducted to achieve more effective drainage of ascitic fluid

In an observational study in 2011, Denver shunts were used as a pleurovenous shunt to move fluid from the pleural cavity to the superior vena cava. [19] This minimally invasive procedure was effective at treating persistent hepatic hydrothorax, but complications, such as fluid overload and coagulopathy, occurred soon after the procedure. [19] Denver shunts differ from peritoneobladder shunts because they connect the peritoneal cavity to venous circulation. When the procedure involves the vascular system, there are severe risks, and complications can occur.

In an observational study in 2022, an Alfapump was used as a subcutaneous device to drain refractory cirrhotic ascites from the peritoneal cavity into the bladder. [20] The Alfapump procedure was found to be effective at reducing the frequency of paracentesis and the volume of ascitic fluid excreted per month. [20] Because simply placing a shunt in the bladder wall to connect the peritoneal cavity with the bladder may not enable natural drainage of ascitic fluid from the peritoneal cavity to the bladder via the shunt, the Alfapump used a powered motor to transport ascitic fluid out of the peritoneal cavity. [20] However, this has the disadvantage of needing to be charged daily, and the subcutaneous implantation procedure can lead to complications such as skin erosion. [20] In addition, because pumps are placed in both the peritoneal cavity and the bladder, there is an even higher risk of complications such as pump displacement or blockage. [21] Finally, owing to the difficulties involved in repeating the procedure, the Alfa pump is not used for malignant ascites, which requires drainage of the exudate that is likely to block the catheter. It can only be used for cirrhotic ascites (transudate). In this regard, we believe that the peritoneobladder shunt could be used for malignant ascites because the procedure could be repeated relatively easily even if catheter obstruction occurrs.

In a case report published in 2024, refractory ascites due to primary intestinal lymphangiectasia was controlled using a peritoneovenous shunt. [22] Using a subcutaneous shunt connected to a one-way pressure valve, the peritoneal cavity was connected to the venous system (internal jugular vein or superior vena cava) to enable continuous drainage of the ascitic fluid. [22] However, since primary intestinal lymphangiectasia is a disease that usually affects children and young adults, this shunt could be difficult to use in patients with general ascites. In addition, because blood vessels are used to drain the ascitic fluid, there are risks of vessel obstruction or thrombus formation. [22]

In a retrospective study in 2024, a peritoneovenous shunt was used to control refractory ascites in patients with decompensated liver cirrhosis. [23] Using a transcutaneous route, the inferior vena cava was connected to the peritoneal cavity

via the right subclavian vein to induce continual drainage of the ascitic fluid. [23] However, the sample size was small, and subjective evaluation of the patients who underwent the procedure was not included. In addition, because blood vessels are used to drain the ascitic fluid, there are risks of vessel obstruction and thrombus formation. [23]

This study is valuable because it is the first to attempt internal fluid drainage without moving the fluid extracorporeally. The focus was on verifying the feasibility of internal drainage using a self-developed peritoneobladder shunt. Additionally, the significance of this study lies in confirming whether the one-way check valve developed for this experiment can effectively prevent reflux from the bladder to the peritoneal cavity.

## Clinical significance of the peritoneobladder shunt procedure

Ascitic fluid in the peritoneal cavity is naturally drained into the bladder through the peritoneobladder shunt, eliminating the need for additional procedures or treatments to reduce the volume of ascitic fluid. This can reduce complications associated with ascites, such as dyspnea, restricted mobility, and pleural effusion. Furthermore, by decreasing the frequency of hospital visits required for repeated paracentesis, the time burden on patients can be alleviated. These points underscore the importance of further research into the internal drainage of ascites and suggest that this device could become a convenient treatment option for patients in the future.

Especially during infectious disease outbreaks such as the COVID-19 pandemic, many patients face significant barriers to accessing medical facilities, and many die because they do not receive necessary treatment in time. Patients who needed periodic ascites drainage often had to visit medical facilities and endure long waits of several days or leave without receiving treatment, causing significant inconvenience. Previously, patients with ascites often experienced dyspnea because of fluid accumulation and had to visit medical facilities each time, and many were unable to receive timely treatment for their breathing difficulties, resulting in the need for mechanical ventilation. The peritoneobladder shunt used in this experiment allows for drainage of ascitic fluid through urine at home after the procedure, reducing the inconvenience of frequent hospital visits. Additionally, continuous drainage of ascites through the peritoneobladder shunt can decrease complications such as dyspnea, preventing the need for mechanical ventilation.

Recently, interest in improving patients' rights has increased, and social activities are emerging as important factors. In this context, since the peritoneobladder shunt is placed inside the body without any external device, it is expected that patients will experience no discomfort during movement. This can be seen as part of a home-based treatment that enables social activities during illness.

## Study limitations

This first study to attempt internal drainage had several limitations and areas for improvement. Two peritoneobladder shunts were placed in swine model 1. However, owing to inexperience with the procedure, one of the peritoneobladder shunts was not properly secured and became dislodged. In contrast, in swine models 2–4 both peritoneobladder shunts were well fixed, which can be attributed to increased familiarity with the procedure. These findings indicate that a skilled and proficient technique is needed for successful shunt placement.

In addition, a quantitative assessment is needed to evaluate how effectively ascitic fluid is drained into the bladder after peritoneobladder shunt placement and whether placing more than two peritoneobladder shunts enhances the drainage effect. In this study, quantifying the amount of ascitic fluid drained was difficult because the swine models tended to bite the Foley catheter, increasing the risk of complications such as bladder rupture. This limitation will be addressed in ongoing follow-up studies.

Since the peritoneobladder shunt was applied to a pig model in this study, it was not possible to assess factors such as post-operative discomfort caused by the device. Although the Piglet Grimace Scale can be used, it is a subjective tool, making accurate evaluation of postoperative discomfort difficult. In future clinical trials, it will be important to assess whether the device causes discomfort in humans.

Owing to the anatomical structure of male pigs, and only female swine models were utilized. Although there are no significant anatomical differences in the bladder between sexes, determining whether differences in other parts of the urinary tract according to sex might affect the procedural outcomes was not possible.

We are currently conducting additional experiments to determine whether the peritoneobladder shunt can remain stably fixed in the bladder wall for more than 7 days. We inserted a peritoneobladder shunt using laparoscopy under general anesthesia; however, this is an invasive procedure. Therefore, when applied to humans, it is essential to explore less invasive methods of insertion. In subsequent follow-up experiments, we investigated less invasive approaches using cystoscopy or nephroscopy instead of laparoscopy. More details on this topic will be discussed in a follow-up manuscript.

Furthermore, it is necessary to verify whether inflammation or infection occurs when the device is inserted into the bladder wall and to consider appropriate treatment options if such complications arise. As the peritoneobladder shunt is a device developed in house by the research team, toxicological testing is needed to verify whether any materials used are harmful to the human body. Toxicity tests on the peritoneobladder shunt are currently underway, and further details will be provided in a follow-up publication.

In this study, we evaluated reflux visually using methylene red; however, this method has limitations in detecting microscopic reflux. For a more accurate assessment of reflux, radiological examinations are needed. We are currently using imaging methods such as angiography to detect any micro reflux.

## Author contributions

**Conceptualization:** Il Hwan Kim, Jung Hyuk Ko, Yong June Lee.

**Data curation:** Byeong Hwa Bak, Jung Hyuk Ko, Ki Won Kim.

**Formal analysis:** Il Hwan Kim, Byeong Hwa Bak, Jung Hyuk Ko, Ki Won Kim.

**Funding acquisition:** Il Hwan Kim.

**Investigation:** Il Hwan Kim, Ki Won Kim, Joo Yeon Kim.

**Methodology:** Il Hwan Kim, Myoung Joo Kang, Ki Won Kim, Joo Yeon Kim, Seok Jae Huh, Cheol Kyu Oh.

**Project administration:** Il Hwan Kim, Ki Won Kim, Jung Hoon Kim, Cheol Kyu Oh.

**Resources:** Byeong Hwa Bak, Ki Won Kim.

**Software:** Byeong Hwa Bak, Myoung Joo Kang, Joo Yeon Kim, Jae Joon Kim, Cheol Kyu Oh.

**Supervision:** Il Hwan Kim, Yong June Lee, Joo Yeon Kim, Jung Hoon Kim, Jae Joon Kim, Seok Jae Huh, Cheol Kyu Oh.

**Validation:** Il Hwan Kim.

**Visualization:** Joo Yeon Kim.

**Writing – original draft:** Byeong Hwa Bak, Jung Hyuk Ko.

**Writing – review & editing:** Il Hwan Kim, Byeong Hwa Bak, Myoung Joo Kang, Jung Hoon Kim, Jae Joon Kim, Seok Jae Huh.

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
