## [Decision Letter · Decision Letter 0]

22 Sep 2025

Dear Dr. Kim,

Thank you for submitting your manuscript to PLOS ONE. After careful consideration, we feel that it has merit but does not fully meet PLOS ONE’s publication criteria as it currently stands. Therefore, we invite you to submit a revised version of the manuscript that addresses the points raised during the review process.

We look forward to receiving your revised manuscript.

Kind regards,

Takehiko Hanaki, MD, PhD

Academic Editor

PLOS ONE

Journal Requirements:

2. To comply with PLOS One submissions requirements, in your Methods section, please provide additional information regarding the experiments involving animals and ensure you have included details on (1) methods of sacrifice, (2) methods of anesthesia and/or analgesia, and (3) efforts to alleviate suffering.

“This research was supported by a grant of the Korea Health Technology R&D Project through the Korea Health Industry Development Institute (KHIDI) funded by the Ministry of Health & Welfare, Republic of Korea (grant number: RS-2024-00407339)”

5. We note that your Data Availability Statement is currently as follows: All relevant data are within the manuscript and in Supporting Information files.

6. Please include a copy of Table 1-3 which you refer to in your text on page 5 and 8.

Reviewers' comments:

Reviewer's Responses to Questions

**Comments to the Author**

1. Is the manuscript technically sound, and do the data support the conclusions?

Reviewer #1: No

Reviewer #2: Partly

Reviewer #3: Partly

2. Has the statistical analysis been performed appropriately and rigorously?

Reviewer #1: N/A

Reviewer #2: N/A

Reviewer #3: N/A

3. Have the authors made all data underlying the findings in their manuscript fully available?

Reviewer #1: Yes

Reviewer #2: Yes

Reviewer #3: No

4. Is the manuscript presented in an intelligible fashion and written in standard English?

Reviewer #1: No

Reviewer #2: Yes

Reviewer #3: Yes

Reviewer #1: This manuscript reports a feasibility study of a peritoneobladder shunt tested in a swine model. The concept is clinically attractive and clearly novel. At the same time, the study is limited by the very small sample size (n=4), the short follow-up period of only seven days, and the lack of quantitative assessment. The conclusions should therefore be presented with more caution.

1. The evaluation of shunt function relied solely on visual confirmation of blue urine and inspection of reflux under laparoscopy. No quantitative data on ascitic fluid drainage, urine output, or intra-abdominal and bladder pressures were provided. Without such data, the evidence for true functional efficacy remains weak.

2. In this device, the direct connection between the bladder and peritoneal cavity raises an important risk of urinary tract infection. Yet no culture data or microbiological evaluation are reported here. This omission should be addressed.

3. When considering eventual human application, the clinical implications of increased urine volume and possible urinary frequency should be carefully discussed. These issues could have a real impact on patient quality of life and should not be underestimated.

4. The only post-mortem assessment was macroscopic confirmation of shunt fixation in the bladder wall. No histological evaluation of the bladder or peritoneum was performed. Microscopic assessment of inflammation, fibrosis, or foreign-body reaction would substantially strengthen the safety evaluation.

5. The study provides no information on longer-term stability or the potential for shunt obstruction. With only seven days of follow-up, it remains entirely unclear whether the device would remain patent in the presence of urine salts, crystal deposition, or biofilm formation. Medium- to long-term experiments are essential before any clinical translation can be seriously contemplated.

Reviewer #2: Malignant ascites is a condition that severely affects patients’ quality of life, often requiring frequent paracentesis. In healthcare settings with limited accessibility, it can lead to serious health problems for patients. Therefore, I have become very interested in animal experiments exploring internal drainage as an alternative approach. However, I have several questions regarding this, and I would like to share my thoughts with you.

1. Even considering the difficulties of animal experiments and their similarities to humans, the sample size is far too small. In addition, the duration of the experiment is too short. From a clinical perspective, situations requiring long-term or even permanent implantation should be considered; therefore, a period of only seven days seems too short to have meaningful implications.

2. In the case of ascites, parameters such as the amount of production and drainage are important, and considering the concept of a shunt, assessing the efficiency of drainage is essential for meaningful interpretation. However, this aspect is not clearly addressed, and relying solely on dye staining for confirmation is unlikely to provide significant meaning.

3. The fact that the study could only be conducted in one sex also represents a significant limitation that cannot be overlooked.

4. If drainage alone is the purpose, it would seem more efficient to permanently place and manage a port directly in the abdomen for drainage.

Reviewer #3: This study reports a feasibility experiment of a peritoneobladder shunt for ascites drainage using a swine model. The work demonstrates that natural drainage into the bladder is technically possible and that one-way valves can prevent reflux. The study is clearly written and includes detailed procedural descriptions. However, several essential elements are missing or require clarification before the manuscript can be considered for publication.

Major points:

•Missing tables: Tables 1–3 are referenced in the text and listed in the Table Legends, but the actual tables are not included in the PDF and no links are provided. Please upload these tables so that reviewers can verify the data.

•Quantitative data: The results are described qualitatively. Please add quantitative or semi-quantitative information where possible (drainage volume, drainage onset time, number of reflux checks, temperature trends).

•Figure presentation: Figures 4–9 are presented as individual images, making it difficult to synthesize the findings. Consider combining representative images or providing a summary figure comparing valve vs. non-valve models and showing before/after placement.

•Results clarity: Present results numerically where appropriate (e.g., 4/4 animals with drainage confirmed). A consolidated results table or schematic summarizing the main findings would improve clarity.

•Discussion balance: Some statements about clinical application are overly optimistic. These should be moderated to emphasize that the findings are limited to a short-term animal model and that further long-term and human studies are required.

•Limitations section: While several limitations are noted, please expand on potential human issues such as urinary frequency, discomfort, and risk of urinary tract infection.

•Animal welfare details: Provide more detail on postoperative monitoring and euthanasia to document compliance with institutional and ARRIVE guidelines.

Minor points:

•Streamline repetitive sentences between Introduction and Discussion.

•Ensure consistent terminology for “peritoneobladder shunt” throughout the text.

**Do you want your identity to be public for this peer review?** For information about this choice, including consent withdrawal, please see our Privacy Policy

Reviewer #1: No

Reviewer #2: No

Reviewer #3: No

---

## [Author Response · Author response to Decision Letter 1]

27 Sep 2025

Reviewer #1

1. The evaluation of shunt function relied solely on visual confirmation of blue urine and inspection of reflux under laparoscopy. No quantitative data on ascitic fluid drainage, urine output, or intra-abdominal and bladder pressures were provided. Without such data, the evidence for true functional efficacy remains weak.

: The purpose of this study was to evaluate the drainage capability through the peritoneobladder shunt. Specifically, we aimed to determine whether ascites could be naturally drained through the shunt when intra-abdominal pressure increases, whether there is any reflux from the bladder to the peritoneal cavity, and whether the shunt remains properly fixed within the bladder. Based on the points you raised, we are planning to address them in a follow-up experiment using a swine model equipped with a neck collar, to investigate whether it allows us to obtain more quantitative data.

2. In this device, the direct connection between the bladder and peritoneal cavity raises an important risk of urinary tract infection. Yet no culture data or microbiological evaluation are reported here. This omission should be addressed.

: The purpose of this study was to evaluate the drainage capability through the shunt. We agree that identifying any potential urinary tract infection is important, as you pointed out. In our ongoing follow-up experiments, we plan to conduct urine culture tests collected from the swine model. Thank you for your valuable feedback.

3. When considering eventual human application, the clinical implications of increased urine volume and possible urinary frequency should be carefully discussed. These issues could have a real impact on patient quality of life and should not be underestimated.

: Yes, it is important to take your point into consideration. In actual clinical trials, we need to evaluate how frequently and how much urine is drained through the shunt. In the ongoing follow-up experiments, we will investigate this by collecting quantitative data from the swine model.

4. The only post-mortem assessment was macroscopic confirmation of shunt fixation in the bladder wall. No histological evaluation of the bladder or peritoneum was performed. Microscopic assessment of inflammation, fibrosis, or foreign-body reaction would substantially strengthen the safety evaluation.

: Yes, it is important to take your comments into consideration. In this experiment, due to budget and laboratory limitations, we were not able to perform a tissue biopsy from the bladder area where the shunt was placed. In the follow-up experiment, we plan to conduct a pathological evaluation using microscopy to address this issue.

5. The study provides no information on longer-term stability or the potential for shunt obstruction. With only seven days of follow-up, it remains entirely unclear whether the device would remain patent in the presence of urine salts, crystal deposition, or biofilm formation. Medium- to long-term experiments are essential before any clinical translation can be seriously contemplated.

: Yes, the point you raised is indeed a limitation of our experiment. Due to budget and laboratory constraints, we were unable to conduct a long-term study in this experiment. We plan to carry out follow-up experiments when longer-term observation beyond 7 days becomes feasible in the future.

Reviewer #2

1. Even considering the difficulties of animal experiments and their similarities to humans, the sample size is far too small. In addition, the duration of the experiment is too short. From a clinical perspective, situations requiring long-term or even permanent implantation should be considered; therefore, a period of only seven days seems too short to have meaningful implications.

: Yes, the point you raised is indeed a limitation of our experiment. In this study, the number of subjects was limited due to budget and laboratory constraints, and we were also unable to conduct long-term experiments for the same reasons. We are planning follow-up studies with an increased number of subjects and longer observation periods of more than 7 days when conditions allow.

2. In the case of ascites, parameters such as the amount of production and drainage are important, and considering the concept of a shunt, assessing the efficiency of drainage is essential for meaningful interpretation. However, this aspect is not clearly addressed, and relying solely on dye staining for confirmation is unlikely to provide significant meaning.

: Yes, it is important to take your comments into consideration. In actual clinical trials, we need to evaluate how frequently and how much urine is drained through the shunt. Although it was difficult to obtain quantitative data from the swine model due to laboratory limitations in this experiment, we plan to collect and analyze quantitative data from the swine model in the ongoing follow-up study.

3. The fact that the study could only be conducted in one sex also represents a significant limitation that cannot be overlooked.

: The swine bladder is very similar to that of humans, which is why it was chosen as the model for this study. However, the anatomical structure of the urinary system differs between male and female pigs, and male pigs are generally not used in urological animal studies for this reason. Therefore, most urological animal experiments are conducted using swine models. In future clinical trials, however, it will be necessary to include both male and female subjects.

4. If drainage alone is the purpose, it would seem more efficient to permanently place and manage a port directly in the abdomen for drainage.

: Thank you for your thoughtful feedback. When a drainage port is permanently placed through the skin into the abdomen, there is a high risk of exposure to external bacteria or viruses, which significantly increases the chance of infection. If the risk of infection is high, it can lead to serious conditions such as peritonitis. A representative example is peritoneal dialysis, where a drainage port is placed on the abdominal surface, and due to the high risk of peritonitis, it is rarely used in Korea. In contrast, the internal drainage method used in this experiment does not come into contact with the external environment, and therefore poses a much lower risk of such infections.

Reviewer #3

Major points:

•Missing tables: Tables 1–3 are referenced in the text and listed in the Table Legends, but the actual tables are not included in the PDF and no links are provided. Please upload these tables so that reviewers can verify the data.

: Yes, I have revised the manuscript to include the table.

•Quantitative data: The results are described qualitatively. Please add quantitative or semi-quantitative information where possible (drainage volume, drainage onset time, number of reflux checks, temperature trends).

: Yes, it is important to take your comments into consideration. Although it was difficult to obtain quantitative data from the swine model due to laboratory limitations in this experiment, we plan to collect and analyze quantitative data from the swine model in the ongoing follow-up study. Regarding the reflux check you mentioned, we confirmed it by visually observing through laparoscopy whether methylene blue refluxed from the bladder into the peritoneal cavity.

•Figure presentation: Figures 4–9 are presented as individual images, making it difficult to synthesize the findings. Consider combining representative images or providing a summary figure comparing valve vs. non-valve models and showing before/after placement.

: To make the manuscript easier to understand, the figures have been labeled as A, B, C, and D, and combined into a single image.

[Fig 4] A Two peritoneobladder shunts placed in the bladder of swine model 1, confirmed laparoscopically B One peritoneobladder shunt confirmed to be situated in the bladder wall of swine model 1 in the dismantling experiment]

[Fig 5] A Two peritoneobladder shunts placed in the bladder of swine model 2 without reflux from the bladder back into the peritoneal cavity, confirmed laparoscopically B Confirmation of two peritoneobladder shunts placed in the bladder wall of swine model 2 in the dismantling experiment C Confirmation of two peritoneobladder shunts placed in the bladder wall of swine model 3 in the dismantling experiment D Confirmation of two peritoneobladder shunts placed in the bladder wall of swine model 4 in the dismantling experiment]

•Results clarity: Present results numerically where appropriate (e.g., 4/4 animals with drainage confirmed). A consolidated results table or schematic summarizing the main findings would improve clarity.

: Yes, the information was organized in the table but was missing, so it has now been included in the manuscript.

Swine model 1 Swine model 2 Swine model 3 Swine model 4

Age / Sex 12 weeks / F 12 weeks / F 12 weeks / F 12 weeks / F

Weight 42 kg 45 kg 42 kg 40 kg

Rearing location Minjeong Livestock Farm (173, Myeongdong-ro, Hallim-myeon, Kimhae-si, Gyeongsangnam-do).

Feed and feeding schedule Swine feed (Purina Lintek Plus 90)

Feeding twice per day, 600 g each time

Rearing environment Temperature 21–26 °C

Humidity 30–70%

Light-dark cycle 12 hrs

Automated water supply

Air ventilation rate 10–15 times/hr

Underlying disease None

[Table 2. Age, sex, weight, feed and feeding schedule, rearing environment, and underlying disease in the swine model]

Swine model peritoneobladder shunt type peritoneobladder shunt placement / count Verification of methylene blue excretion via the Foley catheter Verification of

methylene red

reflux from the bladder to the peritoneal cavity Survival period

1 Check-valves absent Placed / 2 shunts Excreted Reflux observed 7 days

2 Check-valves present Placed / 2 shunts Excreted No reflux observed 7 days

3 Check-valves present Placed / 2 shunts Excreted No reflux observed 7 days

4 Check-valves present Placed / 2 shunts Excreted No reflux observed 7 days

[Table 3. Experimental results for swine models 1–4: peritoneobladder shunt type, verification of peritoneobladder shunt placement, number of peritoneobladder shunts placed, verification of methylene blue excretion via the Foley catheter, verification of methylene red reflux from the bladder to the peritoneal cavity, and survival period]

•Discussion balance: Some statements about clinical application are overly optimistic. These should be moderated to emphasize that the findings are limited to a short-term animal model and that further long-term and human studies are required.

: The purpose of this study was to evaluate the possibility of drainage through the shunt. Specifically, we aimed to determine whether ascitic fluid could be naturally drained through the shunt when intra-abdominal pressure increases, whether reflux occurs, and whether the shunt remains securely fixed in the bladder. Therefore, further studies are necessary, and these points were presented as limitations in the discussion section.

•Limitations section: While several limitations are noted, please expand on potential human issues such as urinary frequency, discomfort, and risk of urinary tract infection.

: Yes, it is important to take your comments into consideration. In actual clinical trials, since ascitic fluid is drained through the urine via the shunt, patients are expected to urinate more frequently than usual. Therefore, it is necessary to consider how often and how much urine is produced, as well as any discomfort experienced. We plan to collect quantitative data on this in ongoing follow-up experiments.

•Animal welfare details: Provide more detail on postoperative monitoring and euthanasia to document compliance with institutional and ARRIVE guidelines.

: We have added information on anesthesia, postoperative monitoring, and euthanasia with reference to the ARRIVE guidelines.

“30 minutes before the start of the procedure, we administered analgesia, a total of two doses of alfaxalone 20 mg and two doses of xylazine 10 mg to the swine model by intramuscularly. At the same time, one dose of tramadol 50 mg was also administered to relieve the pain by intravenously.

After completing the procedure and before transfer to the animal facility, one dose of tramadol 50 mg was also administered to relieve the pain by intravenously. Following transfer to the animal facility, tramadol 50 mg was administered twice daily by intramuscularly.

In dismantling study, 30 minutes before the start of the procedure we administered analgesia, a total of two doses of alfaxalone 20 mg and two doses of xylazine 10 mg to the swine model by intramuscularly. At the same time, one dose of tramadol 50 mg was also administered to relieve the pain by intravenously.”

“4) Verification of performance and specimen analysis after the procedure

After evaluating short term (7 days) survival, we performed a dismantling experiment to verify that the peritoneobladder shunt was properly fixed in the bladder.

(1) Experiment preparation

① 30 minutes before the start of the procedure we administered analgesia, the swine model was anesthetized using two doses of alfaxalone 20 mg and two doses of xylazine 10 mg per animal by intramuscularly. At the same time, one dose of tramadol 50 mg was also administered to relieve the pain by intravenously.

② After confirming that the swine model was anesthetized the swine model was secured to the operating table in the Trendelenburg position.

(3) The surgical site and surrounding area were covered with 7.5% povidone sterile drapes.

(4) The rest of the swine model was covered with sterilized drapes to avoid contamination.

(2) Dismantling experiment

① After confirming that the swine model was anesthetized, KCl 40 mg was administered by intravenously and cardiac arrest was verified.

② The bladder was isolated and the peritoneobladder shunt was confirmed to still be well placed within the bladder.”

“After the swine model returned to the animal facility, during the 7 days observation period, we visually confirmed that the methylene blue dye was excreted with urine and stained the animal’s skin, such as on the legs and hips. Throughout this period, core temperature measured three times a day that showed no signs of fever or infection and the swine model’s food intake remained consistent with previous levels, feeding twice per day, 600 g each time. Upon dismantling experiments after 7 days, one peritoneobladder shunt remained properly fixed in the bladder wall, whereas the other had become detached and was found inside the bladder [Fig 4B]”

“After the swine model returned to the animal facility, during the 7 days observation period, we visually confirmed that the methylene blue dye was excreted with urine and stained the animal’s skin, such as on the legs and hips. Throughout this period, core temperature measured three times a day that showed no signs of fever or infection, and the swine model’s food intake remained consistent with previous levels, feeding twice per day, 600 g each time. Upon dismantling experiments after 7 days, two peritoneobladder shunt remained properly fixed in the bladder wall, [Fig 5B], [Fig 5C], [Fig 5D].”

Minor points:

•Streamline repetitive sentences between Introduction and Discussion.

: Yes, I have incorporated your suggestions and edited the manuscript through a proofreading service..

•Ensure consistent terminology for “peritoneobladder shunt” throughout the text.

: Yes, I have made the revisions based on your comments.

---

## [Editor Report · Decision Letter 1]

4 Nov 2025

Feasibility study of a novel technique for treating refractory ascites using a peritoneobladder shunt in a swine model

PONE-D-25-41418R1

Dear Dr. Kim,

We’re pleased to inform you that your manuscript has been judged scientifically suitable for publication and will be formally accepted for publication once it meets all outstanding technical requirements.

Kind regards,

Takehiko Hanaki, MD, PhD

Academic Editor

PLOS ONE

Additional Editor Comments (optional):

Although this study evaluates only short-term outcomes and several medium- to long-term concerns remain, it presents valuable preliminary findings.

I have determined that the manuscript has sufficient merit to warrant acceptance.
---

## [Editor Report · Acceptance letter]

PONE-D-25-41418R1

PLOS One

Dear Dr. Kim,

I'm pleased to inform you that your manuscript has been deemed suitable for publication in PLOS One. Congratulations! Your manuscript is now being handed over to our production team.

Kind regards,

on behalf of

Dr. Takehiko Hanaki

Academic Editor

PLOS One